# Low Levels of Few Micronutrients May Impact COVID-19 Disease Progression: An Observational Study on the First Wave

**DOI:** 10.3390/metabo11090565

**Published:** 2021-08-24

**Authors:** Teresa-Maria Tomasa-Irriguible, Lara Bielsa-Berrocal, Luisa Bordejé-Laguna, Cristina Tural-Llàcher, Jaume Barallat, Josep-Maria Manresa-Domínguez, Pere Torán-Monserrat

**Affiliations:** 1Intensive Care Unit, University Hospital Germans Trias i Pujol, 08916 Badalona, Spain; larabielsa@gmail.com (L.B.-B.); luisabordeje@gmail.com (L.B.-L.); 2Internal Medicine Department, University Hospital Germans Trias i Pujol, 08916 Badalona, Spain; ctural.germanstrias@gencat.cat; 3Biochemical Department, University Hospital Germans Trias i Pujol, 08916 Badalona, Spain; jbarallat.germanstrias@gencat.cat; 4North Metropolitan Research Support Unit, Jordi Gol i Gurina Foundation Institute for Research in Primary Health Care (IDIAPJGol), 08303 Mataró, Spain; jmanresa@idiapjgol.info (J.-M.M.-D.); ptoran.bnm.ics@gencat.cat (P.T.-M.)

**Keywords:** vitamins, trace elements, micronutrients, coronavirus, SARS-CoV-2, COVID-19, critical illness, outcome

## Abstract

We report an observational study performed between March and May 2020 in a Spanish university hospital during the SARS-CoV-2 pandemic. The main objective was to analyse the association between the levels of micronutrients in severe COVID-19 patients and their outcome. Adult patients with a positive polymerase-chain-reaction (PCR) for SARS-CoV-2 in the nasopharyngeal swab or in tracheal aspirate culture in the case of intubation were included. Micronutrient data were obtained from plasma analysis of a standard nutritional assessment performed within the first 24 h of hospital admission. Vitamins A, B_6_, C and E were analysed with HPLC methods; 25-OH-vitamin D by immunoassay and zinc by colorimetric measurements. One hundred and twenty patients were included. We found that 74.2% patients had low levels of zinc (normal levels >84 µg/dL) with a mean value of 63.5 (SD 13.5); 71.7% patients had low levels of vitamin A (normal levels >0.3 mg/L) with a mean value of 0.17 (SD 0.06); 42.5% patients had low levels of vitamin B_6_ (normal levels >3.6 ng/mL) with a mean value of 2.2 (SD 0.9); 100% patients had low levels of vitamin C (normal levels >0.4 mg/dL) with a mean value of 0.14 (SD 0.05); 74.3% patients had low values of vitamin D (normal levels >20 ng/mL) with mean value of 11.4 (SD 4.3); but only 5.8% of patients had low levels of vitamin E (normal levels >5 mg/L) with a mean value of 3.95 (SD 0.87). The variables associated with the need for ICU admission were low levels of zinc (standard error 0.566, 95% CI 0.086 to 0.790, *p* = 0.017), low levels of vitamin A (standard error 0.582, 95% CI 0.061 to 0.594, *p* = 0.004), age over 65 (standard error 0.018, 95% CI 0.917 to 0.985, *p* = 0.005) and male gender (standard error 0.458, 95% CI 1.004 to 6.040, *p* = 0.049). The only variable that was independently associated with the need for orotracheal intubation was low levels of vitamin A (standard error 0.58, 95% CI 0.042 to 0.405, *p* = 0.000). Conclusions: Low levels of vitamin A and zinc are associated with a greater need for admission to the ICU and orotracheal intubation. Patients older than 65 years had higher mortality. Randomized clinical trials are needed to examine whether micronutrient supplementation could be beneficial as an adjunctive treatment in COVID-19.

## 1. Introduction

The severe acute respiratory syndrome Coronavirus-2 (SARS-CoV-2) represents an emerging global threat that is severely depleting global health capacity. As of 21 July 2021, 191,281,182 confirmed cumulative cases of COVID-19 have been reported globally, including 4,112,538 deaths [1]. Until worldwide vaccination is achieved, it is extremely important to study and develop appropriate treatments for the prevention and treatment of COVID-19. Safe and inexpensive interventions with a strong biological basis should be the priority for experimental use in the current context of the pandemic [2]. A previous study had shown that despite the expected efficacy of lopinavir/ritonavir, IFN-β-1a and remdesivir, these drugs are unlikely to have a significant effect on viral kinetics when given as monotherapy after the onset of symptoms [3]. Moreover, a recent study has shown that treatments that were given in the first wave as lopinavir/ritonavir and hydroxychloroquine and the combination therapy worsened outcomes compared to no COVID-19 antiviral therapy, reducing organ support-free days and survival among critically ill patients [4]. Therefore, there is an unmet medical need to provide new tools to reduce the risk of infection, as well as the morbidity and mortality of SARS-CoV-2, while awaiting the development and effectiveness of the new vaccines and new prophylactic drugs [5]. The reasons for the wide variation in severity of COVID-19 are unknown. Genetic, race, gender, anthropometric index, previous medications, among others, are currently being studied. Nutritional status is an aspect that is also under investigation, as well as the relationship that may exist between micronutrients deficiency and immunity. Micronutrients have previously been studied and their deficiency has been associated with a greater number of infections and worse outcomes [6,7] and it is postulated that their supplementation may improve the incidence and severity of infections [8,9]. Vitamin D status seems to have a linear association with seasonal infections and lung function [10]. Vitamin D supplementation seems to protect against acute respiratory tract infection and patients who are very vitamin D deficient experience the most benefit [11]. In addition, vitamin D deficiency is linked to poorer seroconversion to the H3N2 and B strains of the seasonal influenza vaccine [12]. Zinc has also been related to improved immune response to vaccination [13,14]. Zinc inhibits coronavirus RNA polymerase activity in vitro and zinc ionophores block the replication of these viruses in cell culture [15]. A relationship may exist between micronutrients deficiency and the severity of COVID as well. Regarding COVID-19 infection, low plasma 25(OH) vitamin D is associated with increased risk of this SARS-CoV-2 infection, resulting an independent risk factor for COVID-19 infection and hospitalization [16]. Blood vitamin D status can determine the risk of being infected with COVID-19 and its severity [17]. Regarding zinc, patients with COVID-19 have low zinc levels in comparison to healthy controls, and zinc deficiency patients develop more complications and had prolonged hospitalization [18]. Zinc low levels are also a predictive factor for critical illness [19]. Not only zinc, but selenium deficiency is also associated with mortality risk from COVID-19 [20]. Finally, a relationship may exist between micronutrient supplementation and the recovery from COVID-19. Vitamin D supplementation is associated to better survival in elderly COVID-19 patients [21] and treatment with cholecalciferol booster therapy is associated with a reduced risk of mortality in acute in-patients admitted with COVID-19 [22]. Zinc may also improve outcomes in hospitalized COVID-19 patients [23]. 

In the light of existing literature, we conducted an observational study with the aim of assess whether low levels of micronutrients are associated with outcome clinical setting of severe COVID-19. The main objective was to analyse the association between the levels of micronutrients and the need for orotracheal intubation in patients with COVID-19. And the secondary objectives were: To analyse the association between the of micronutrients and the need for ICU admission, ICU and hospital stay, mortality, need for invasive mechanical ventilation, development of multi-organ failure, need for renal replacement therapy and development of other common complications in COVID-19, such as new-onset cardiac dysfunction and thromboembolic disease (deep vein thrombosis and pulmonary thromboembolism).

## 2. Results

A total of 120 patients were included in the study in the first wave. The most relevant results of the study are described point by point below.

### 2.1. Characteristics of the Patients 

Most of the patients included in the study were male (63.3%), the mean age was 58.7 years, and their mean body mass index was 29.7 kg/m^2^. The most frequent symptoms at disease onset were fever (86%), dyspnoea (68%) and cough (64%). All patients met criteria for moderate (32.5) or severe (67.5) ARDS criteria. Even though all patients were admitted for severe COVID-19, only 50 patients were admitted to the ICU. So that, at that time, ICU admission criteria was orotracheal intubation or extreme work of breathing. The other 70 patients remained in ward despite their severe condition. Patients’ characteristics are summarized in Table 1. 

### 2.2. Micronutrients Levels

Most of the patients included had low blood levels of all micronutrients tested but for vitamin E. We found that 74.2% patients had low levels of zinc with a mean value of 63.5 (SD 13.5) (normal levels >84 µg/dL); 71.7% patients had low levels of vitamin A with a mean value of 0.17 (SD 0.06) (normal levels >0.3 mg/L); 42.5% patients had low levels of vitamin B6 with a mean value of 2.2 (SD 0.9) (normal levels >3.6 ng/mL); 100% patients had low levels of vitamin C with a mean value of 0.14 (SD 0.05) (normal levels >0.4 mg/dL); 74.3% patients had low values of vitamin D with mean value of 11.4 (SD 4.3) (normal levels >20 ng/mL); but only 5.8% of patients had low levels of vitamin E with a mean value of 3.95 (SD 0.87) (normal levels >5 mg/L). Other micronutrients’ levels are summarized in Table 2. 

### 2.3. Association between Low Micronutrient Levels and Clinical Variables 

Low levels of vitamin A were associated with male sex (69% vs. 45%, *p* = 0.02), with the need for ICU admission (62.1% vs. 20.7%; *p* = 0.048), the orotracheal intubation rate (92.3% vs. 7.7%; *p* = 0.000), the need for prone position (93.6% vs. 6.4%; *p* = 0.000), the need for norepinephrine (92.9% vs. 7.1%; *p* = 0.001) and a higher rate of respiratory bacterial superinfection (92.6% vs. 7.4%; *p* = 0.016).

Low levels of zinc were associated with older age (63 vs. 57, *p* = 0.016) and were also associated with a higher ICU admission rate (61.8% vs. 29%; *p* = 0.002), higher rate of orotracheal intubation (87.5% vs. 12.5%; *p* = 0.002), higher need for prone position (86% vs. 14%; *p* = 0.013), higher need for norepinephrine (89.1% vs. 10.9%; *p* = 0.001) and a higher rate of respiratory bacterial superinfection (90% vs. 10%; *p* = 0.023). Other micronutrients’ associations are summarized in Table 3 and Table 4. 

Finally, we did not find statistically significant differences with the levels of micronutrients and the other variables as new-onset cardiac dysfunction, the need for dobutamine, the presence of deep vein thrombosis, nor pulmonary embolism.

The multivariable logistic regression models show that the variables independently associated with the need for ICU admission were low levels of zinc (OR 3.84, 95% CI 1.27 to 11.65, *p* = 0.017), low levels of vitamin A (OR 5.26, 95% CI 1.68 to 16.46, *p* = 0.004), age (OR 0.95, 95% CI 0.92 to 0.98, *p* = 0.005) and male sex (OR 2.46, 95% CI 1.00 to 6.04, *p* = 0.049) (Table 5). On the other hand, the variables associated with the need for orotracheal intubation was the low levels of vitamin A (OR: 6.66, 95% CI 2.10 to 21.15, *p* = 0.001) and male sex (OR 2.57, 95% CI 1.09 to 6.06, *p* = 0.031).

### 2.4. Association between Inflammatory Parameters and Clinical Variables 

Higher IL-6 and ferritin levels were associated with male sex, and older patients had lower levels of prealbumin. Patients with elevated plasma D-dimer values upon admission also had a higher ICU admission rate (59.6% vs. 25%; *p* = 0.011) and a higher orotracheal intubation rate (2.5% vs. 7.5%; *p* = 0.046). In addition, high ferritin values at admission were also associated with the need for ICU admission (62.7% vs. 9.1%; *p* = 0.000), the orotracheal intubation rate (97.8% vs. 2.2%; *p* = 0.000), the need for prone position (92.9% vs. 7.1%; *p* = 0.001) and a higher rate of bacterial respiratory superinfection (95.7% vs. 4.3%; *p* = 0.017). Patients with low values of prealbumin and high values of ferritin had lower values of vitamin A. The other inflammatory variables were not associated with lower levels of the other micronutrients. Likewise, high levels of IL-6 at admission were also associated with a greater need for prone position (100% vs. 0%; *p* = 0.043). Other association between inflammatory parameters and clinical variables are summarized in Table 3.

### 2.5. Association between Variables and Mortality

The deceased patients had lower levels of vitamins A, B6 and zinc, although without statistical significance. In addition, patients with low prealbumin levels and high levels of CRP, and IL-6 had higher mortality, although no statistically significant differences were found. Finally, only the age was associated with mortality (OR: 1.16, 95% CI 1.08 to 1.24, *p* < 0.001).

## 3. Discussion

In this sample of 120 patients with severe COVID-19 disease, low plasma levels of micronutrients were found to be the general trend in most of them. Moreover, low plasma levels of zinc and vitamin A were associated with an increased need for ICU admission and orotracheal intubation. Whether, these findings are the cause or the consequence of the disease, and, whether these findings are related to the patient’s inflammatory state or a state of micronutrient deficiency remains to elucidate. 

Regarding the low levels of vitamin A and its relationship with the male sex, it could be argued that perhaps men intake of foods containing vitamin A is lower, or likewise that men have a higher metabolic expenditure of vitamin A when they become seriously ill. And regarding the relationship between low zinc levels and age, it could also be argued that perhaps zinc intake is lower in older people.

In an ecological study conducted in non-hospitalized patients, the authors observed that vitamin and mineral requirements are not satisfied by the diet in the European Union and that it could be a limiting factor in the immune response. Spain showed the worst data in relation to insufficient vitamin D and vitamin A intake, and it was the country with the highest incidence of COVID-19 and the second in mortality at the time the data analysis was performed [24]. Reduced intake and malabsorption of nutrients can lead to a vitamin deficiency, which can facilitate the development of infections. These findings have been described in other series of septic patients [25,26]. De Pascale et al., in a study with 107 subjects, found that 93.5% were deficient in vitamin D [27]. Similarly, Carr et al., in a cohort of 24 patients, reported a vitamin C deficiency in more than 40%. In addition, those patients over age 65 had a significant vitamin deficiency without necessarily presenting sepsis. Furthermore, malnutrition has a negative impact on the immune system, consequently increasing the risk of sepsis [28]. Additionally, the study conducted by Galmés et al. showed a genetic risk assessment of the European population, which was carried out by adding the total number of risk alleles present in the individual sample. The study found that Spanish, Italian and English people had a higher genetic risk for vitamins A, B_12_ and D deficiencies [24]. It should be noted that the countries with the worst intake profile for these micronutrients correspond to those that have received the cruelest blow from the COVID-19 pandemic. Another study has shown that vitamin A may have anti-SARS-CoV-2 effects [29]. These effects are modulated by genes, including MAPK1, IL10 and EGFR, among others. A study performed by Choi et al., reported that the patients with EGFR gene mutations were more likely to acquire nosocomial pneumonia [30]. Several studies have observed that other micronutrients like vitamin C and D have antiviral effect as well as a positive effect on immunity [31,32,33]. In addition, a pilot study highlighted that the administration of a high dose of calcifediol (25-hydroxyvitamin D) significantly reduced, in hospitalized patients with COVID-19, the need for ICU admission [34]. Currently, very few treatments have been established for SARS-CoV-2, some of which have shown limited efficacy in clinical practice. In the study carried out by the RECOVERY group, dexamethasone has shown to reduce mortality in those patients with COVID-19 who required mechanical ventilation and oxygen therapy [35]. The REACT Working Group in a recent meta-analysis of patients hospitalized for COVID-19, showed that the administration of IL-6 antagonists is associated with lower mortality [36]. Nonetheless, patients are still being admitted to the ICU after dexamethasone and tocilizumab combination treatment and treatments are still needed to prevent hospitalizations for COVID-19. 

Both vitamin A and zinc could provide strong and beneficial pharmacological activity in the treatment of COVID-19 through their antiviral, antioxidant and anti-inflammatory effects. The present study found that low plasma levels of zinc and vitamin A are associated with an increased need for ICU admission and orotracheal intubation in patients with severe COVID-19 pneumonia. As we could not measure the micronutrient status at baseline without disease, we don’t know if this is a matter of deficiency of micronutrients or it is a matter of a redistribution of these micronutrients because of inflammation, or both. In the present study, patients with low values of prealbumin and high values of ferritin and IL-6 had lower values of vitamin A. However, these inflammatory variables were not associated with lower levels of the other micronutrients including Zinc. Regarding vitamin A and inflammatory parameters, it is possible that these factors could alter vitamin A levels, so that vitamin A status was not compromised, or that this inflammatory status could consume vitamin A in excess, and that vitamin A levels could be truly low. There are some studies that have linked the systemic inflammatory response (SIR) with low levels of micronutrients. Duncan et al., showed that zinc, selenium and vitamins A, B6, C and D were lower as C-reactive protein (CRP) increased [37]. They concluded that interpretation of plasma micronutrients can be made only with knowledge of the degree of inflammatory response. Otherwise, in a recent study conducted by Gonçalves et al. carried out in critically ill patients infected by SARS-CoV-2 admitted to an ICU for severe ADRS (pO2/FiO2 ≤ 100 mmHg), they found that the prevalence of low zinc levels among these patients was 79.6% (95% CI, 74.7–84%). They also found that there was a statistical association between low plasma levels of zinc and severe ADRS, but they didn’t find a statistically significant difference between critically ill patients with normal or low zinc levels and CPR concentrations on the first and third day of admission in the ICU [38]. Nevertheless, although inflammation can alter plasma levels of micronutrients, as well as the alteration of its binding proteins, we must bear in mind that in Spain there is a genetic predisposition to have low levels of vitamin A and zinc, in addition to an inadequate intake of these micronutrients [24]. Other authors have found similar results as well and concluded that vitamin A severely reduced plasma levels in COVID-19 patients are significantly associated with ARDS and mortality [39]. Though plasma levels may not accurately represent the real status of such micronutrients, it does not mean there is or there is not a deficiency status. It only means that currently there is no method good enough to properly measure the status of these micronutrients in real time. It would be interesting to be able to individualize the supplementation knowing the plasma levels of micronutrients in real time. Besides, supplementation with micronutrients may or may not enhance the faculty to fight the virus, or enhance the therapeutic efficacy of current clinical antivirals and immunotherapy or vaccines for the treatment of COVID-19. 

### Limitations

As this was an observational study, we cannot establish a relationship between the need for orotracheal intubation and ICU with low levels of vitamin A and zinc and we only can generate hypotheses. The sample is a bit small, and new follow-up studies should be performed with larger samples to help confirm these results. A second limitation is that other vitamins and metals such as iron, copper or selenium were not tested. In addition, the nutritional analysis was carried out once at hospital admission, so we were not able to follow the evolution of their micronutrient plasma levels during the COVID-19 disease. Finally, the data on nutritional status that we could recorded was limited, we were unable to obtain the exact weight loss in the days prior to admission, as well as the loss of appetite and the intake in the last days.

Randomized clinical trials are needed to assess if micronutrient supplementation could be beneficial in improving the outcome of COVID-19 patients. Today’s date there have been published few randomized clinical trials, and some of them with negative results [40,41]. As of today, it is not known if supplementation with a single micronutrient is sufficient to improve the prognosis of COVID-19. 

## 4. Materials and Methods

This was an observational epidemiological study conducted in a referral university hospital during the pandemic between March and May 2020. In this section the study design, the inclusion criteria, the diagnostic tests and lab analyses, and finally the statistical analysis are described. 

### 4.1. Design/Participants

A cross-sectional descriptive study to explore whether low levels of micronutrients predict a worse prognosis in patients with severe COVID-19 pneumonia admitted in a single tertiary referral hospital. The patients included were those admitted to the Internal Medicine and Intensive Care departments of the Germans Trias i Pujol University Hospital in Badalona, Spain, in the first wave. The inclusion criteria were adult patients who met ARDS criteria according to the Berlin definition [42], who were diagnosed of COVID-19 with a positive polymerase-chain-reaction (PCR) for COVID-19 in the nasopharyngeal swab or in tracheal aspirate culture in the case of intubation admitted because of severe COVID-19 disease. The exclusion criteria were the age of the patients younger than 18 years or that patients were admitted for a disease other than severe COVID-19.

### 4.2. Materials

Different real-time PCR diagnostic test were used for the detection of SARS-COV-2: Diasorin (Gene 5 and ORF), GenXpert (Gene E and N2), Aliplex (Gene E, R and N) and recommended tubes available commercially by the European Centre for Disease Prevention and Control (Gene N1 and N2). 

### 4.3. Data Source and Variables

Micronutrient data were obtained from serum analysis of a standard nutritional assessment performed at hospital admission within the first 24 h, exceptionally within 48 h. The explored micronutrients were vitamin A (retinol), vitamin B_6_ (pyridoxine), vitamin C (ascorbate), vitamin D (25-hydroxyvitamin D), vitamin E (α-tocopherol) and zinc. The vitamin C sample refers only to the 55 critically ill patients admitted at the ICU, because of the difficulties in obtaining reliable samples in ward. Vitamins A, B_6_, C and E were analysed with HPLC methods; 25-OH-Vitamin D by immunoassay and zinc by colorimetric measurements. Other nutritional analytical parameters collected were body mass index (BMI), prealbumin and albumin. The immune response variables collected in plasma were D-dimer, interleukin-6 (IL-6), ferritin, C-reactive protein (CRP). 

We also collected demographic data such as age, gender, and the presence of chronic renal failure, in addition to recording current renal function, the need for orotracheal intubation, ICU admission, length of stay both in ICU and hospital, mortality, Sequential Organ Failure Assessment (SOFA) score at ICU admission, need for prone position, respiratory bacterial superinfection, vasoactive and inotropic drugs, new-onset of cardiac dysfunction diagnosed by echocardiography, deep vein thrombosis (DVT) diagnosed by ultrasound of the limbs, presence of pulmonary embolism (PE) diagnosed by echocardiography or CT scan or pulmonary gammagraphy, likewise acute renal failure and need for intermittent dialysis and continuous renal replacement therapy (CRRT). 

### 4.4. Statistical Analysis

The qualitative variables are summarized with their absolute and relative frequencies, and the quantitative variables with their mean and standard deviation. For the comparison between proportions we performed Pearson’s Chi Square tests and t Student tests for continuous variables. Multivariate logistic regression models were adjusted to determine the association between low micronutrient levels and outcomes (admission to the intensive care unit [ICU] and orotracheal intubation [OTI]) SSB consumption and lifestyle and sociodemographic variables. In the first bivariate regression models, we identified variables that were individually associated with the two outcome variables at *p* < 0.20. These variables were then included in saturated multivariate models. The final models were adjusted for the level of significance and biological plausibility. Statistical significance was set at *p* < 0.05 (two-tailed). Data were analyzed with SPSS (IBM Corp. Released 2017. IBM SPSS Statistics for Windows, Version 25.0. Armonk, NY, USA).

## 5. Conclusions

The present study shows that most patients with severe COVID-19 pneumonia have low levels of multiple micronutrients. Low levels of vitamin A and zinc, age over 65 years and male gender are factors associated with the necessity of ICU admission. Patients with low levels of vitamin A have a greater need for orotracheal intubation, a definitive factor that determines ICU admission. These results may be of interest because they involve factors that are easily modified by the clinician, such as supplementation with micronutrients if necessary. Since it is an observational study, no definitive conclusions can be drawn, so clinical trials should be conducted to assess the efficacy of micronutrient supplementation in the progression of COVID-19 disease.

## Figures and Tables

**Table 1 metabolites-11-00565-t001:** Characteristics of patients included in the sample, management procedures and treatments of patients included in the sample.

Characteristic	Results
Age (years), Mean (SD)	58.74 (13.9)
Female	43 (35.8)
Male	77 (64.2)
Weight (kg), Mean (SD)	89.4 (32.8)
Heigh (cm), Mean (SD)	162.3 (23)
BMI (kg/m^2^), Mean (SD)	29.7 (12)
Headache, Cases, n (%)	43 (35.8)
Anosmia/Ageusia, Cases, n (%)	7 (5.8)
Myalgia, Cases, n (%)	33 (27.5)
Diarrhoea, Cases, n (%)	35 (29.2)
Fever, Cases, n (%)	103 (85.8)
Cough, Cases, n (%)	77 (64.1)
Dyspnoea, Cases, n (%)	81 (67.5)
Chest pain, Cases, n (%)	24 (20)
Smoker, Cases, n (%)	9 (7.5)
Alcoholism, Cases, n (%)	6 (5)
Mellitus diabetes, Cases, n (%)	20 (16.7)
Arterial hypertension, Cases, n (%)	39 (32.5)
Chronic obstructive pulmonary disease, Cases, n (%)	6 (5)
Asthma, Cases, n (%)	15 (12.5)
Obesity, Cases, n (%)	46 (38.3)
Sleep apnoea-hypopnea syndrome, Cases, n (%)	12 (10)
Dyslipidaemia, Cases, n (%)	47 (39.2)
Ischemic heart disease, Cases, n (%)	22 (18.3)
Heart failure, Cases, n (%)	2 (1.7)
Hydroxychloroquine, Cases, n (%)	63 (52.5)
Chloroquine, Cases, n (%)	23 (19.2)
Darunavir, Cases, n (%)	19 (15.8)
Lopinavir/Ritonavir, Cases, n (%)	50 (41.7)
Remdesivir, Cases, n (%)	1 (0.8)
Interferon, Cases, n (%)	33 (27.5)
Tocilizumab, Cases, n (%)	20 (16.7)
Steroids, Cases, n (%)	36 (30)
Ceftriaxone, Cases, n (%)	61 (51)
Azithromycin, Cases, n (%)	27 (22.5)
Flu vaccinated, Cases, n (%)	39 (32.5)
SDRA criteria:	
Mild ARDS [PaO(2)/FiO2 ratio = 200–300]	0
Moderate ARDS [PaO(2)/FiO2 ratio = 100–200]	39 (32.5%)
Severe ARDS [PaO(2)/FiO2 ratio < 100]	81 (67.5%)
ICU procedures:	
Invasive mechanical ventilation	42 (84)
Non-invasive mechanical ventilation	1 (2)
High-flow oxygen therapy	23 (46)
Prone position	26 (52)
Tracheostomy	16 (32)
Extracorporeal membrane oxygenation	2 (4)
Length of stay:	
LOS of all patients admitted to the hospital, Mean (SD)	19.8 (17.3)
LOS of patients not admitted to the ICU, Mean (SD)	12.9 (8.3)
LOS of patients admitted to the ICU, Mean (SD)	18 (13.5)
Total LOS of patients admitted to the ICU, Mean (SD)	29.7 (21.4)
Total hospital deaths	27 (22.5)

BMI: body mass index; SDRA: Acute Respiratory Distress Syndrome; LOS: length of stay; ICU: intensive care unit.

**Table 2 metabolites-11-00565-t002:** Relevant data of patients based on their micronutrient levels.

	VIT.A (mg/L)		VIT. B6 (ng/mL)		VIT.C (mcg/dL)		VIT.D (ng/mL)		VIT.E (mg/L)		ZINC (µg/dL)	
Laboratory reference values	Normal [0.3–0.6]	Low [<0.3]	Normal [3.6–18]	Low [<3.6]	Normal [0.4–2]	Low [<0.4]	Normal [20–150]	Low [<20]	Normal [5–18]	Low [<5]	Normal [84–159]	Low [<84]
Number of patients ^1^, n (%)	34 (28.3)	86 (71.7)	69 (57.5)	51 (42.5)	0	55 (100)	27 (25.7)	78 (74.3)	113 (94.2)	7 (5.8)	31 (25.8)	89 (74.2)
Sex (male), n (%)	13 (44.8)	60 * (69)	43 (68.3)	30 (57.7)	5 (71.4)	36 (75)	40 (57.1)	25 (71.4)	71 (62.8)	6 (85.7)	19 (61.3)	58 (65.2)
Age (years), mean (SD)	57.8 (13.4)	63 (13.3)	60.1 (14)	63.6 (12.9)	58.6 (7.3)	59.1 (12.1)	63.1 (13.7)	59.7 (13.7)	61.5 (13.3)	61.3 (16)	56.5 (13)	63.2 * (13.2)
Mortality, n (%)	5 (17.2)	25 (29.4)	10 (16.1)	17 (33.3)	2 (28.6)	15 (31.9)	7 (28)	11 (31.4)	28 (25.7)	2 (28.6)	5 (16.1)	26 (29.9)
ICU, n (%) admission	6 (20.7)	54 * (62.1)	38 (60.3)	22 (42.3)	7 (100)	48 (100)	15 (55.6)	18 (51.4)	58 (52.3)	5 (71.4)	9 (29)	55 * (61.8)
NAD, n (%)	3 (7.1)	39 * (92.9)	26 (56.5)	16 (34.8)	6 (15%)	34 (85%)	9 (25%)	17 (47.2)	41 (89.1)	4 (8.7)	5 (10.9)	41 * (89.1)
DBT, n (%)	1 (16.7)	5 (83.3)	3 (42.9)	3 (42.9)	1 (16.7)	5 (83.3)	1 (25)	3 (75)	6 (85.7)	1 (14.3)	0	7 (100)
DVT, n (%)	1 (12.5)	7 (87.5)	5 (55.6)	3 (33.3)	3 (60)	2 (40)	2 (25)	4 (50)	8 (88.9)	1 (11.1)	2 (22.2)	7 (77.8)
PE, n (%)	0	4 (100)	1 (25)	2 (50)	1 (33.3)	2 (66.7)	1 (50)	1 (50)	3 (75)	1 (25)	1 (25)	3 (75)
HF, n (%)	1 (20%)	4 (80)	2 (40)	2 (40)	1 (20)	4 (80)	0	2 (100)	4 (80)	1 (20)	1 (20)	4 (80)
AKI, n (%)	5 (17.2)	24 (82.8)	19 (61.3)	9 (29)	4 (18.2)	18 (81.8)	6 (22.2)	11 (40.7)	28 (90.3)	2 (6.5)	6 (19.4)	25 (80.6)
ID, n (%)	0	3 (100)	2 (50)	2 (50)	1 (50)	1 (50)	0	3 * (100)	4 (100)	0	0	4 (100)
CRRT, n (%)	0	4 (100)	1 (25)	1 (25)	1 (33.3)	2 (66.7)	0	2 (100)	3 (75)	1 (25)	0	4 (100)
OTI, n (%)	4 (7.7)	48 * (92.3)	34 (60.7)	18 (32.1)	7 (14.3)	42 (85.7)	11 (26.2)	17 (40.5)	51 (91.1)	4 (7.1)	7 (12.5)	49 * (87.5)
PP, n (%)	3 (6.4)	44 * (93.6)	29 (58)	18 (36)	7 (17.5)	33 (82.5)	12 (28.6)	17 (40.5)	45 (90)	4 (8)	7 (14)	43 * (86)
BRS, n (%)	2 (7.4)	25 * (92.6)	18 (60)	11 (36.7)	4 (17.4)	19 (82.6)	5 (23.8)	8 (38.1)	29 (96.7)	1 (3.3)	3 (10)	27 * (90)

^1^ Number of patients means the number of patients with normal or low values of the micronutrient. AKI: Acute Kidney Injury; BRS: Bacterial Respiratory Superinfection; CRRT: Continuous Renal Replacement Therapy; DBT: Dobutamine; DVT: Deep Venous Thrombosis; HF: Heart Failure; ID: Intermittent Dialysis; IL(6): Interleukin-6; NAD: Noradrenaline; OTI: Orotracheal Intubation; PE: Pulmonary Embolism; PP: Prone Position. * *p* < 0.05.

**Table 3 metabolites-11-00565-t003:** Relevant data of the patients based on their inflammatory parameters.

	PREALBUMIN mg/dL		CRP mg/L		D-DIMER ng/mL		IL-6 pg/mL		FERRITIN ng/mL	
Laboratory reference values	Normal [20–40]	Low [<20]	Normal [0–5]	High [>5]	Normal [0–500]	High [>500]	Normal [0–6.4]	High [>6.4]	Normal [30–400]	High [>400]
Number of Patients ^1^, n (%)	9 (7.6)	110 (92.4)	0	119 (100)	16 (14.5)	94 (85.5)	5 (4.6)	104 (95.4)	22 (22.7)	75 (77.3)
Sex (male), n (%)	8 (80)	68 (62.4)	0	76 (63.9)	13 (81.39)	58 (61.7)	1 (20)	68 * (65.4)	8 (36.4)	56 * (74.7)
Age (years old)	55.1 (6.9)	62.1 * (13.8)	0	62.1 (13.8)	63.4 (15.4)	60.6 (12.8)	56.6 (7)	61.2 (13.7)	13.3 (17.1)	61.2 (12.1)
Low Vitamin A levels, n (%)	0	86 * (80.4)	0	86 (74.8)	13 (81.3)	68 (74.7)	0	78* (78)	12 (54.5)	58 * (80.6)
Low Vitamin B6 levels, n (%)	4 (40)	48 (46.2)	0	52 (45.2)	9 (56.3)	40 (44.4)	2 (40)	46 (46)	11 (52.4)	35 (47.9)
Low Vitamin C levels, n (%)	6 (100)	42 (85.7)	0	47 (87)	3 (75)	41 (87.2)	2 (100)	42 ( 85.4)	2 (100)	34 (82.9)
Low Vitamin D levels, n (%)	4 (40)	31 (33)	0	35 (33.3)	4 (30.8)	28 (33.7)	2 (40)	30 (32.6)	7 (31.8)	22 (34.4)
Low Vitamin E levels, n (%)	0	7 (6.4)	0	7 (5.9)	1 (6.3)	6 (6.4)	0	7 (6.7)	0	7 (9.3)
Low Zinc levels, n (%)	6 (60)	83 (76.1)	0	89 (74.8)	10 (62.5)	71 (75.5)	3 (60)	78 (75)	14 (63.6)	58 (77.3)
Mortality, n (%)	1 (11.1)	30 (28)	0	30 (25.6)	3 (18.8)	24 (26.1)	0	26 (25.5)	6 (27.3)	19 (25.7)
ICU Admission n (%)	5 (55.6)	57 (52.3)	0	63 (52.9)	4 (25)	56 * (59.6)	2 (40)	57 (54.8)	2 (9.1)	47 * (62.7)
NAD, n (%)	2 (4.4)	42 (93.3)	0	45 (100)	3 (7)	40 (93)	0	41 (100)	1 (2.8)	35 * (97.2)
DBT, n (%)	0	6 (85.7)	0	7 (100)	0	5 (100)	0	5 (100)	1 (16.7)	5 (83.3)
DVT, n (%)	0	9 (100)	0	9 (100)	0	9 (100)	0	9 (100)	0	8 (100)
PE, n (%)	0	4 (100)	0	4 (100)	0	4 (100)	0	3 (100)	0	4 (100)
HF, n (%)	0	5 (100)	0	5 (100)	0	4 (100)	0	3 (100)	1 (20)	4 (80)
AKI, n (%)	2 (6.7)	27 (90)	0	31 (100)	4 (15.4)	22 (84.6)	1 (3.8)	25 (96.2)	2 (8)	23* (92)
ID, n (%)	0	4 (100)	0	4 (100)	0	4 (100)	0	3 (100)	0	4 (100)
CRRT, (%)	0	4 (100)	0	4 (100)	0	4 (100)	0	3 (100)	0	4 (100)
OTI, n (%)	3 (5.5)	51 (92.7)	0	55 (100)	4 (7.5)	49 * (92.5)	1 (2)	50 (98)	1 (2.2)	44 * (97.8)
PP, n (%)	3 (6.1)	45 (91.8)	0	50 (100)	4 (8.3)	44 (91.7)	0	48 * (100)	3 (7.1)	39 * (92.9)
BRS, n (%)	2 (6.9)	26 (89.7)	0	30 (100)	4 (14.3)	24 (85.7)	1 (3.7)	26 (96.3)	1 (4.3)	22 * (95.7)

^1^ Number of patients means the number of patients with normal or low values of their inflammatory status. AKI: Acute Kidney Injury; BRS: Bacterial Respiratory Superinfection; CRP: C—reactive protein; CRRT: Continuous Renal Replacement Therapy; DBT: Dobutamine; DVT: Deep Venous Thrombosis; HF: Heart Failure; ID: Intermittent Dialysis; IL-6: Interleukin-6; NAD: Noradrenaline; OTI: Orotracheal Intubation; PE: Pulmonary Embolism; PP: Prone Position. * *p* < 0.05.

**Table 4 metabolites-11-00565-t004:** Results of the bivariate logistic regression between the low levels of each micronutrients and outcomes (ICU and OTI).

Outcome	Factor	Beta	OR(CI95%)	*p*
ICU	Sex (Men)	1.027	2.79 (1.29–6.04)	0.009
	Age	−0.025	0.98 (0.95–1.00)	0.076
	Vit A (<0.3 mg/L)	1.836	6.27 (2.31–17.01)	0.000
	Vit B6 (<3.6 ng/mL)	−0.729	0.48 (0.23–1.02)	0.056
	Vit C (<0.4 mcg/mL)	---		NV
	Vit D (<20 ng/mL)	−0.229	1.26 (0.56–2.83)	0.581
	Vit E (<5 mg/L)	−0.828	2.29 (0.43–12.29)	0.334
	Zinc (<84 mcg/L)	1.375	3.95 (1.63–9.59)	0.002
OTI	Sex (Men)	1.237	3.44 (1.54–7.70)	0.003
	Age	−0.008	0.99 (0.97–1.02)	0.575
	Vit A (<0.3 mg/L)	2.040	7.69 (2.47–23.98)	0.000
	Vit B6 (<3.6 ng/mL)	−0.795	0.45 (0.21–0.96)	0.039
	Vit C (<0.4 mcg/mL)	---		NV
	Vit D (<20 ng/mL)	−0.531	1.70 (0.75–3.87)	0.207
	Vit E (<5 mg/L)	−0.447	1.56 (0.33–7.31)	0.570
	Zinc (<84 mcg/L)	1.435	4.20 (1.64–10.75)	0.003

**Table 5 metabolites-11-00565-t005:** Summary of the multivariate logistic regression analysis with its initial saturated models and resulting final models, regarding the outcome (ICU and OTI).

Outcome				
**UCI**	**Saturated Model**	**Beta**	**OR(CI95%)**	*p*
	Sex (man)	0.695	2.00 (0.80–5.04)	0.140
	Age	−0.050	0.95 (0.92–0.99)	0.009
	Vit A (<0.3 mg/L)	1.830	6.23 (1.82–21.39)	0.004
	Vit B6 (<3.6 ng/mL)	−0.570	0.57 (0.24–1.34)	0.196
	Zinc (<84 mcg/L)	1.248	3.48 (1.09–11.11)	0.035
	Constant			
	**Final Model**	**Beta**	**OR(CI95%)**	*p*
	Sex (man)	0.901	2.46 (1.00–6.04)	0.049
	Age	−0.051	0.95 (0.92–0.98)	0.005
	Vit A (<0.3 mg/L)	1.661	5.26 (1.68–16.46)	0.004
	Zinc (<84 mcg/L)	1.346	3.84 (1.27–11.65)	0.017
	Constant			
**OTI**	**Saturated Model**	**Beta**	**OR(CI95%)**	*p*
	Sex (man)	0.886	2.42 (0.96–6.13)	0.061
	Age	−0.020	0.98 (0.95–1.01)	0.243
	Vit A (<0.3 mg/L)	1.900	6.68 (1.70–26.25)	0.006
	Vit B6 (<3.6 ng/mL)	−0.654	0.52 (0.22–1.24)	0.139
	Zinc (<84 mcg/L)	1.149	3.16 (0.96–10.33)	0.058
	Constant			
	**Final Model**	**Beta**	**OR(CI95%)**	*p*
	Sex (man)	0.943	2.57 (1.09–6.06)	0.031
	Vit A (<0.3 mg/L)	1.897	6.66 (2.10–21.15)	0.001
	Constant			

## Data Availability

The data presented in this study are available on request from the corresponding author. The data are not publicly available due to privacy restrictions.

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
