# Peer review of "Low Levels of Few Micronutrients May Impact COVID-19 Disease Progression: An Observational Study on the First Wave"

_metabolites, 2021, doi:10.3390/metabo11090565_

Round 1
Reviewer 1 Report
The manuscript titled "Low Micronutrient Levels may Impact the Need for ICU Admission of COVID-19 Patients: An Observational Study in the First Wave." submitted to Metabolites, describes a very important and actual topic related to pandemic situation and COVID-19 patients. The authors properly designed study and proposed to analyse levels of micronutrients especially in patients with severe COVID-19 pneumonia. Blood samples were collected, in total 120 patients were included and authors proposed few qualitative variables. The authors presented and discussed in the manuscript nutritional level and basically authors decided find relation between micronutrients low level and the severity of COVID. The study seems to be properly designed and authors input a lot of effort to reach valuable results and proper conclusions.
Regarding structure of manuscript and content: Abstract and introduction is well written divided into subsections. Methods and Materials are very well organize. Results and Discussion are presented very clearly and describe all experiments very well. Tables and figures are presented correctly.
I think that paper can be accepted for publication on MEtabolites without revisions.
Author Response
Please see attachment below.

Reviewer 2 Report
This manuscript described a study of micronutrient level in COVID-19 patient and concluded that most patient with severe symptom had low level of vitamin A and zinc. In addition, the author concluded low level vitamin A and zinc, age over 65, and male gender are the factors associated with necessity ICU admission. This study is intersecting that related COVID-19 patient micronutrient level to ICU admission t. And point out a potential study about micronutrients administration in COVID-19 therapy could be necessary. Although the authors showed an interesting study topic and results, a larger number of study sample should be considered for a more solid conclusion.
Comments:
- The mean age of studied patient was 58.74. what was the age range of studied patients? Did the author most focus on patient in or older than middle age? If the studied patient age is in a specific range, all the conclusions should specify the applied age range.
- Will region factor influence studied patient’s micronutrient level? Such as the influence of dietary habit.
- In line 144-146, the author mentioned “Spain showed the worst data in relation to insufficient vitamin D and vitamin A intake, and it was the country with the highest incidence of COVID-19 and the second in mortality at the time the data analysis was performed”. How to ascribe severe COVID-19 situation to insufficient vitamin D and vitamin A intake directly? How to exclude other reasons?
- In line 172-174, the author mentioned “Both vitamin A and zinc can provide strong and beneficial pharmacological activity in the treatment of COVID-19 through their antiviral, antioxidant and anti-inflammatory effects”. Please provide references to support it.
Author Response
Reviewer’s comments:
- The mean age of studied patient was 58.74. what was the age range of studied patients?
|
Age (years), Mean (SD) |
58.74 (13.9) |
- Did the author most focus on patient in or older than middle age? If the studied patient age is in a specific range, all the conclusions should specify the applied age range
We did not focus on any particular age, we studied patients as they arrived at the hospital in the first wave
- Will region factor influence studied patient’s micronutrient level? Such as the influence of dietary habit.
It is certainly possible that the region factor may influence the micronutrient level of the patients, as well as the dietary habit. Unfortunately, this data was not collected in the first wave micronutrient study. It would be very interesting to study these issues in future studies.
- The paragraph in line 144-146 “Spain presented the worst data in relation to insufficient intake of vitamin D and vitamin A, and was the country with the highest incidence of COVID-19 and the second in mortality at the time of the analysis of data was carried out ”, it is not from our study, it was a study by Galmés et al. that we found very interesting. We do not believe that Galmés et al. directly attributed a serious situation of COVID-19 to an insufficient intake of vitamin D and vitamin A, they simply state that such a situation coexisted. And, of course, other causes cannot be excluded.
Galmés S, Serra F, Palou A. (2020). Current state of evidence: influence of nutritional and nutrigenetic factors on immunity in the COVID-19 pandemic framework. Nutrients. 12,2738. https://doi.org/10.3390/nu12092738
- In line 172-174, the author mentioned “Both vitamin A and zinc can provide strong and beneficial pharmacological activity in the treatment of COVID-19 through their antiviral, antioxidant and anti-inflammatory effects”. Please provide references to support it :
You are right. I forgot to put the reference. The reference are number 7 and 15.

Round 2
Reviewer 2 Report
The authors have revised the manuscript based on last round comments. Most comments have been addressed. However, I didn't find addition document for the point to point answer/explanation.